# Water Temperature and Hydrological Modelling in the Context of Environmental Flows and Future Climate Change: Case Study of the Wilmot River (Canada)

Christian Charron [1], André St-Hilaire [1,2,*], Taha B.M.J. Ouarda [1] and Michael R. van den Heuvel [2,3]

1   Institut National de la Recherche Scientifique, Centre Eau Terre Environnement,
    Québec, QC G1K9A9, Canada; christian.charron@ete.inrs.ca (C.C.); Taha.ouarda@ete.inrs.ca (T.B.M.J.O.)
2   Canadian Rivers Institute, University of New Brunswick, Fredericton, NB E3B 5A3, Canada;
    mheuvel@upei.ca
3   Department of Biology, University of Prince Edward Island, Charlottetown, PE C1A 4P3, Canada
*   Correspondence: andre.st-hilaire@ete.inrs.ca

**Abstract:** Simulation of surface water flow and temperature under a non-stationary, anthropogenically impacted climate is critical for water resource decision makers, especially in the context of environmental flow determination. Two climate change scenarios were employed to predict streamflow and temperature: RCP 8.5, the most pessimistic with regards to climate change, and RCP 4.5, a more optimistic scenario where greenhouse gas emissions peak in 2040. Two periods, 2018–2050 and 2051–2100, were also evaluated. In Canada, a number of modelling studies have shown that many regions will likely be faced with higher winter flow and lower summer flows. The CEQUEAU hydrological and water temperature model was calibrated and validated for the Wilmot River, Canada, using historic data for flow and temperature. Total annual precipitation in the region was found to remain stable under RCP 4.5 and increase over time under RCP 8.5. Median stream flow was expected to increase over present levels in the low flow months of August and September. However, increased climate variability led to higher numbers of periodic extreme low flow events and little change to the frequency of extreme high flow events. The effective increase in water temperature was four-fold greater in winter with an approximate mean difference of 4 °C, while the change was only 1 °C in summer. Overall implications for native coldwater fishes and water abstraction are not severe, except for the potential for more variability, and hence periodic extreme low flow/high temperature events.

**Keywords:** environmental flow; water temperature; hydrological model; climate change; hydrological forecasting

## 1. Introduction

Determining environmental flows (e-flows) is one of the challenges facing water resources managers. E-flows were defined by Arthington et al. [1] as "*the quantity, timing, and quality of freshwater flows and levels necessary to sustain aquatic ecosystems which, in turn, support human cultures, economies, sustainable livelihoods, and well-being*". This definition provides an indication that prescribing e-flows can be a complex endeavor. Many methods exist to reach a decision of the value(s) of e-flow(s) for a river. They include hydrologic methods that are typically similar to the approach described in the seminal work of Tennant [2]. Habitat models have also been used in many instances to define e-flows that maintain proper physical conditions for certain fish, invertebrate or plant species (e.g., [3]). More holistic approaches include the Building Blocks Methodology [4] and the Ecological Limits of Hydrological Alteration (ELOHA) framework [5]. These holistic approaches attempt to establish links between hydrology, ecosystem functions and human needs [6].

However, in many jurisdictions, e-flows are established by simplifying this process to a single definition of a specific threshold of minimum flow (or maximum water abstraction)

that must remain in the river (or be available for withdrawal). These e-flow statistics are often based on the distribution of historical flows for a given river. For instance, Tennant [2] concluded that an adequate e-flow should maintain at least 30% of mean annual flow (MAF) for rivers with salmonid populations in the U.S. In New Zealand, two hydrological approaches have been used: a percentage (30–75%) of the 1 in 5 years low flow, and a flow that equaled or exceeded 96% of the time [7]. In Britain, the flow duration curve is used to determine a minimum cut-off flow corresponding, also to the flow exceeded 95% of the time (Q95, [8]).

In Canada, where our case study was conducted, e-flow prescriptions vary across provincial jurisdictions. In the Eastern part of the country, Caissie, Caissie and El-Jabi [9] and El-Jabi and Caissie [10] compared a number of hydrological e-flow metrics. They indicated that low flow quantiles of 7-day duration and return periods of 2 (7Q2) and 10 (7Q10) years were deemed insufficient for a number of rivers in the region. However, they mentioned that using a percentage (70%) of the median discharge (70Q50) value of a low flow month (e.g., August in this region) as an e-flow minimum threshold provided better fish habitat conditions than the 7Q2 and 7Q10. In the province of Prince Edward Island (PEI), where the drainage basin used in our case study is located, 70Q50 of monthly mean discharge is used to define e-flows.

While it remains appealing to prescribe e-flows using simple hydrological metrics such as the ones described herein, many ecologists find it too simplistic. Olden and Naiman [11] suggested that another key variable that should be included in e-flow determination is river temperature. Although it is relatively simple to measure, river temperature is often overlooked despite the pivotal role it plays for all aquatic fauna and flora. Moreover, river temperature is known to be affected by anthropogenic impacts such as agriculture [12], flow regulation [13] and climate change [14]. A number of recent studies, including some in Eastern Canada [14,15] and elsewhere [16] have shown that the increase in air temperature and potential lower summer flows in rivers may result in significant increases in summer river temperature. In PEI, where most rivers host stenotherm fish such as salmonids and where angling is an important activity, the repercussions could be important.

Given the non-stationary context of climate change, future discharge in PEI rivers will likely shift and may include more extreme values, thereby modifying the empirical (or modelled) non-exceedance probabilities of low flows. In this context, deterministic hydrological models are useful tools that can be implemented to simulate future hydrological scenarios.

The objective of the present study is to investigate current and possible future hydrological and thermal conditions during low flow months on a small agricultural watershed in the context of re-visiting environmental flows guidelines. This is done by implementing the CEQUEAU hydrological and water temperature model [17,18] to simulate both flow and water temperature on one PEI drainage basin: the Wilmot River. The CEQUEAU hydrological and water temperature model was calibrated and validated using historic data from the period 1972–2012 for flow and 2013–2017 for temperature. This model has been successfully used in similar previous studies to simulate future hydrological and thermal scenarios [19,20]. However, no published studies have focused on Prince Edward Island. Specific objectives include: (1) providing a first calibration of the model on historical flows and water temperature and assess its performance; (2) generating hydrological and water temperature scenarios for the 2050/2100 horizons; (3) compare e-flow metrics from historical time series and future scenarios with the associated water temperatures for the first time in this region.

## 2. Materials and Methods

### 2.1. Trend Analyses

Trend analyses were performed on historical precipitation time series and on future precipitation scenarios (described hereafter) in order to determine if either RCP 4.5 and/or RCP 8.5 climate change scenario projected a statistically significant change in precipitation, as precipitation is a major component driving stream flow. The statistical test used

was the Mann–Kendall test for monotonic trends [21]. In addition, to evaluate potential changes in the annual patterns of rainfall, monthly precipitation was examined with a non-parametric Analysis of Similarities (ANOSIM) using PRIMER V.7 software [22] with a calibration/validation period as the categorical factor. This was followed by a similarity percentages breakdown (SIMPER) analysis, which evaluates the individual contribution per month to the similarity using a Bray–Curtis similarity matrix (similarity that varies between 1 when samples have the same composition and 0 when they are entirely different).

### 2.2. CEQUEAU Model

CEQUEAU is a deterministic semi-distributed hydrological and water temperature model [23]. The hydrological module of CEQUEAU considers the physical characteristics of the watershed by decomposing it in hydrological units of surfaces of equal area (termed "whole squares"). For each whole square, altitude, percentage of forest area and percentage of area covered by lakes and wetlands must be defined. Water routing is defined initially by partitioning whole squares into a maximum of four so-called "partial square" according to the water divides. This subdivision of up to four partial squares in each whole square allows us to define water routing at the proper scale.

Routing is performed by apportioning the available water originating from surface runoff, interflow (flow from the unsaturated soil horizon indicated as the upper-zone in Figure 1) and base-flow (flow from groundwater, indicated as the lower-zone in Figure 1) proportionally to partial square areas and identifying the receiving partial square downstream.

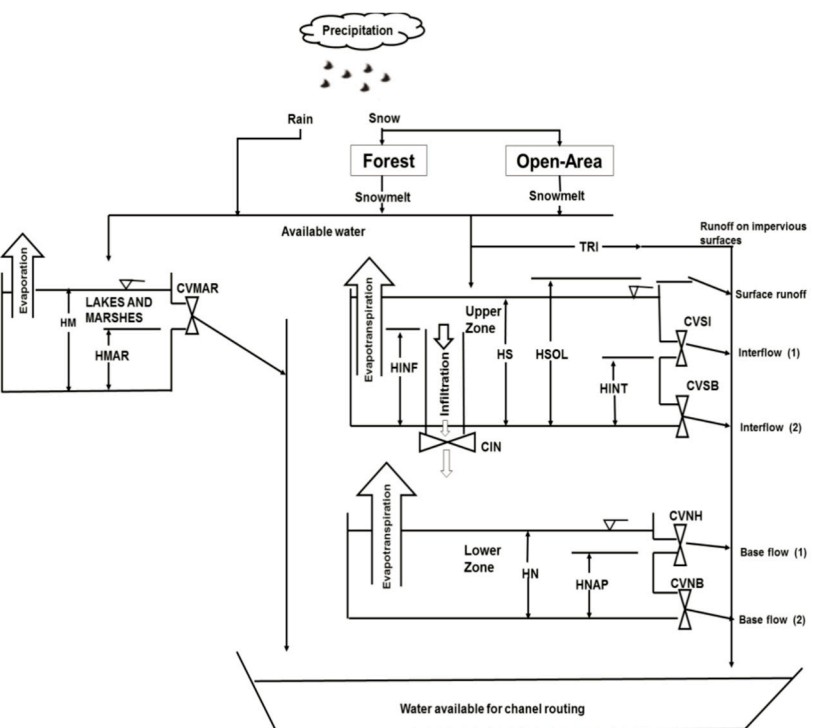

**Figure 1.** Production function vertical water routing in CEQUEAU (adapted from [17,24]).

CEQUEAU is composed of two main functions, the production function responsible for quantifying the vertical flow of water and the transfer function, which calculates a routing coefficient responsible for quantifying upstream–downstream water advection. The production function is modelled by a series of reservoirs, two of which are connected, as they estimate the soil infiltration (upper-zone) and soil water storage (lower-zone). The third one represents the contribution of lakes and wetlands in terms of storage and runoff (Figure 1; [24]). For each whole square at time $t$, the resulting production function is:

$$Q(t_n) = P(t_n) - ETP(t_n) + (HU(t_n) - HU(t_{n-1})) + (HL(t_n) - HL(t_{n-1})) \quad (1)$$

where $Q$ (mm) is the total runoff, $P$ (mm) is the measured rainfall or the estimated snowmelt, $ETP$ (mm) is the evapotranspiration, $HU$ (mm) is the water accumulated in the upper reservoir, $HL$ (mm) is the water accumulated in the lower reservoir, $t_n$ is the time at daily time step $n$ for $n = 1, \ldots, N$ and $N$ is the total number of time steps (days). ETP is calculated using the Thorntwaite method [23]. Snowmelt is estimated using the degree-day method of the U.S. Army Corps of Engineers [23].

The transfer function, which allows available water to be routed downstream from one partial square to another to simulate discharge at each time step is given by:

$$XKT_i = 1 - \exp\left( -\frac{EXXKT \cdot RMA3_i}{\max(Sl_i, Slac_i)} \cdot \frac{100}{CEKM2} \right) \tag{2}$$

where $XKT_i$ is the routing coefficient of the $i$th partial square, $EXXKT$ is the fitting parameter, $RMA3_i$ is the area of the basin upstream of the $i$th partial square (km$^2$), $Sl_i$ is the area of the total surface water upstream of the $i$th partial square, $Slac_i$ (km$^2$) is the area of surface water on the $i$th partial square, and $CEKM2$ (km$^2$) is the area of the whole square.

At the same time step, the hydrological simulation results are fed to the thermal module in addition to other meteorological data (solar radiation, wind speed, air vapor pressure and cloud cover). The thermal component of the model is based on a simple heat budget calculated for each hydrological unit. Change in temperature is calculated using Equation (3):

$$T_w = \frac{H}{V \cdot C} \tag{3}$$

where $H$ represents the total enthalpy (Energy in MJ) of the thermodynamic system; $V$, the volume of water (m$^3$) and $C$ the heat capacity of water (C = 4.187 MJ m$^{-3}$ $^{\circ}$C$^{-1}$). The heat budget calculated to obtain $H$ accounts for the following fluxes:

- Short wave radiation (measured);
- Long wave incoming and backscattered radiation (calculated using the Stefan–Boltzmann equation; [23]);
- Evapotranspiration (latent heat; calculated as a function of the difference between saturated vapor pressure and water vapor pressure in the air; [23]);
- Convection (sensible heat; estimated from an empirical equation based on the Bowen Ratio; [23]);
- Upstream and downstream advection;
- Local heat advection (from runoff, interflow and groundwater inputs).

For both hydrological and thermal modules, model validation is performed using a split sample method. The historical time series was split in two sub-samples; one part was used for calibration and the other for verification of the calibrated model. The parameters of the hydrological model were optimized using Monte Carlo Simulations with Taboo Search [25]. Model performances were evaluated with the Bias (Equation (4)), the root-mean-square error (RMSE, Equation (5)) and the Kling and Gupta efficiency (KGE, Equation (6)).

$$Bias = \frac{1}{n} \sum_{t=1}^{n} (\hat{y}_t - y_t) \tag{4}$$

$$RMSE = \sqrt{\frac{1}{n} \sum_{t=1}^{n} (\hat{y}_t - y_t)^2} \tag{5}$$

$$KGE = 1 - \sqrt{(\rho - 1)^2 + (\alpha - 1)^2 + (\beta - 1)^2} \tag{6}$$

In Equations (4)–(6), $n$ is sample size, $y_t$ is the observed value of the variable of interest (flow or temperature), $\hat{y}_t$ is the simulated value and $t$ is the time step. $\alpha = \sigma_s / \sigma_o$ is the ratio between the standard deviation of the simulated and the observed values; $\beta = \mu_s / \mu_o$ is the ratio between the means of the simulations and the observations and $\rho$ is the linear correlation between the simulated and observed values.

### 2.3. Study Site, Model Implementation and Calibration

The Wilmot River watershed is in southwestern PEI (Figure 2) and drains an area of 49 km². Approximately 80% of this area is dedicated to agriculture. This drainage area was divided in whole squares of 0.25 km² each. Meteorological inputs for the hydrologic module (daily maximum and minimum air temperature, as well as total daily precipitation) were extracted from the ANUSPLIN database [26]. Location of ANUSPLIN grid points used to interpolate meteorological inputs on each whole square are shown in red in Figure 2. The thermal module requires additional meteorological inputs (solar radiation, vapor pressure, cloud cover, wind speed). Those variables are not available in the ANUSPLIN database. The National Center for Environmental Protection (NCEP) North American Regional Reanalyses (NARR) were therefore used (available online: https://www.ncep.noaa.gov/, accessed on 31 May 2017). NARR reanalyses are produced for all North America on a grid with a resolution of 0.33° × 0.33°. The data from the NARR grid point closest to the drainage basin (Figure 2) were used for calibration.

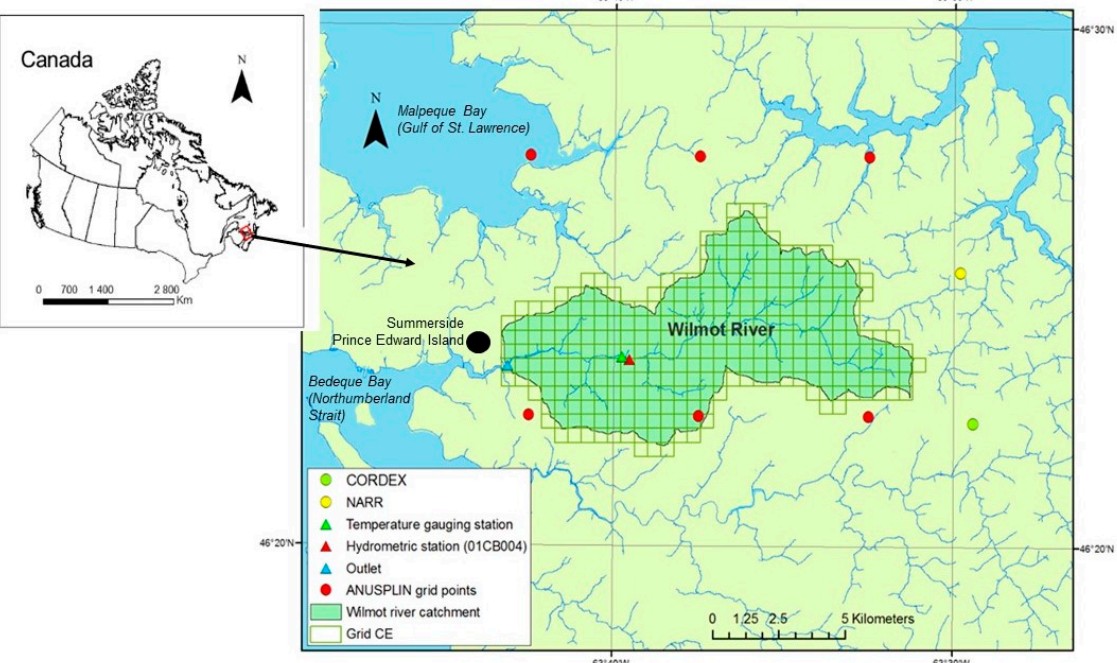

**Figure 2.** Wilmot River drainage basin with the CEQUEAU whole square grid. ANUSPLIN (**red**), CORDEX (**green**) and NARR (**yellow**) grid point locations are also shown. Temperature monitoring and hydrometric station locations are also shown. The city of Summerside and adjacent estuaries are indicated, the larger bodies of water into which those bays flow is indicated in parenthesis.

Calibration of a hydrologic model requires an objective function that must be optimized (minimized) by adjusting the model parameters. The Kling Gupta Efficiency (KGE) criterion was used in this study as the objective function (Equation (6)). As stated in the previous section, model parameters are estimated automatically using the method presented by Larabi et al. [25]. Calibration of the hydrologic module was done first using a split sample approach. The calibration period used was 1992–2012 and the validation period 1972–1991.

Calibration of the thermal module could not be done using the split sample approach because of the relative paucity of available water temperature data on the Wilmot River. For this project, time series of daily mean temperatures from 2013 to 2017 were used. A jackknife approach was used for calibration, whereby the model is calibrated on a four-year sub-sample of the original dataset (e.g., 2014–2017) and validated on the sub-sample of the year that was left out (e.g., 2013). This process is repeated for each validation year from 2013 to 2017. An overall optimal model is finally obtained by taking the average value of

each parameter computed in the calibrated models. This model is then used to produce the thermal simulations for the climate change scenarios.

### 2.4. Climate Change Scenarios

To investigate possible future changes in the hydrological and thermal regime of the Wilmot River, two climate change scenarios were used. These scenarios are generated using climate models that are run with different greenhouse gas emission scenarios (called Representative Concentration Pathways or RCP) and provide climatic time series of meteorological variables used as inputs to the CEQUEAU model. These scenarios are merely realizations of possible future outcomes according to more or less optimistic predictions of GHG (Green House Gas) emissions. Two such scenarios were used in this work. RCP 4.5 is a relatively optimistic scenario in which GHG emissions peak around 2040, followed by a decline. With RCP 8.5, the more pessimistic scenario, GHG emissions continue to rise during the entire 21st century. Meteorological time series for the 2100 horizon for these two scenarios were obtained from the Coordinated Regional Climate Downscaling Experiment (CORDEX; Available online, https://www.cordex.org/, accessed on 1 June 2017). Their grid resolution is set to $0.44° \times 0.44°$ (approximately $40 \times 40$ km). Climate projections for each RCP scenarios were generated with the CCCma-CanESM2 global climate model, the SMHI-RCA4 regional climate model and r1i1p1 ensemble members. Again, data from the grid point closest to the drainage basin (Figure 2) were used as meteorological inputs for both the hydrological and thermal modules to generate future flow and water temperature scenarios.

## 3. Results

### 3.1. Climate Change Precipitation Scenarios

Annual rainfall patterns were compared between the calibration (1992–2012) and validation periods (1972–1991) to ensure stability between intervals and over the 40-year period (Figure 3A). There was no statistically significant change in annual precipitation ($K = 0.022$, $p = 0.983$), nor was mean total precipitation different between calibration and validation periods despite three drought years (2001–2002, 2012) occurring in the calibration data set.

While the calibration period showed more variability in annual pattern, there was no statistically significant change in the seasonal (monthly) rainfall patterns between the calibration and validation datasets.

Precipitation-year regressions based on the climate scenarios show no significant change in annual precipitation between 2018 and 2100 using RCP 4.5 (Figure 3B, slope = 0.039, $K = 1.71$, $p = 0.09$). However, the RCP 8.5 scenario show a significant predicted increase in annual precipitation between 2018 and 2100 (Figure 3B, slope = 2.45, $K = 4.66$, $p < 0.001$). To determine if the annual monthly distribution of precipitation was predicted to change, monthly precipitation patterns were evaluated using ANOSIM with 1972–2012 observed precipitation data, RCP 4.5 2018–2050, RCP 4.5 2051–2100, RCP 8.5 2018–2050, and RCP 8.5 2051–2100. All climate change scenarios showed significantly different patterns than the observed precipitation data ($p < 0.01$), though none of the climate scenarios precipitation patterns differed from each other. A SIMPER analysis was conducted to determine which months contributed most to the dissimilarity. In three of the comparisons between observed and scenario data, July and August were the months that contributed most to the dissimilarity, in the fourth case it was July and October, with August being the third highest contributor to dissimilarity. Precipitation future scenarios overestimated summer precipitation as compared to data observed between 1972 and 2012. To confirm this, an ANOVA (Analysis of variance) was conducted on August total precipitation for the same five periods/scenarios. Observed August precipitation was significantly lower than August precipitation of all four climate scenarios/periods ($p < 0.05$).

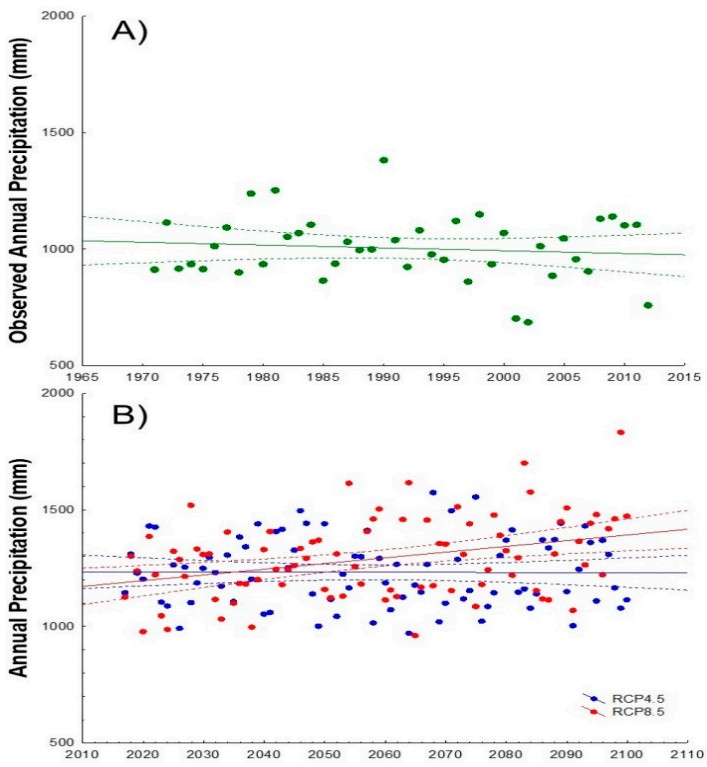

**Figure 3.** Annual precipitation for observed data from 1972–2012 (**A**), and modelled data for the two climate change scenarios (**B**). Linear trends (solid lines) and associated confidence intervals at a level of 95% (dashed lines) are indicated.

### 3.2. Flow Model Calibration and Validation

Flow model performance criteria are summarized for both the calibration and validation periods in Table 1. KGE values are adequate (i.e., >50%) for both periods, albeit surprisingly weaker during calibration (58%) than during validation (65%). Figure 4 presents the interannual daily means of observed and simulated flows for the calibration and validation periods. The spring flood appears to be more adequately simulated during the validation period. Also, during validation, the early floods that occur in late March were more adequately simulated than during calibration. The validation period is characterized by a systematic overestimation of fall and early winter discharge. This overestimation may be due to biases in the ANUSPLIN meteorological inputs [26]. The flood recession curve is well simulated during both periods. Low flow simulations are relatively good, except for the minimum flows that occur during the month of September for which there is a systematic (albeit weak) underestimation by the model.

**Table 1.** Performance criteria for the simulated flows.

|  | Calibration Sample (1992–2012) | Validation Sample (1972–1991) |
|---|---|---|
| KGE (%) | 57.63 | 64.98 |
| Bias (m³/s) | −0.041 | 0.031 |
| RMSE (m³/s) | 1.038 | 0.973 |

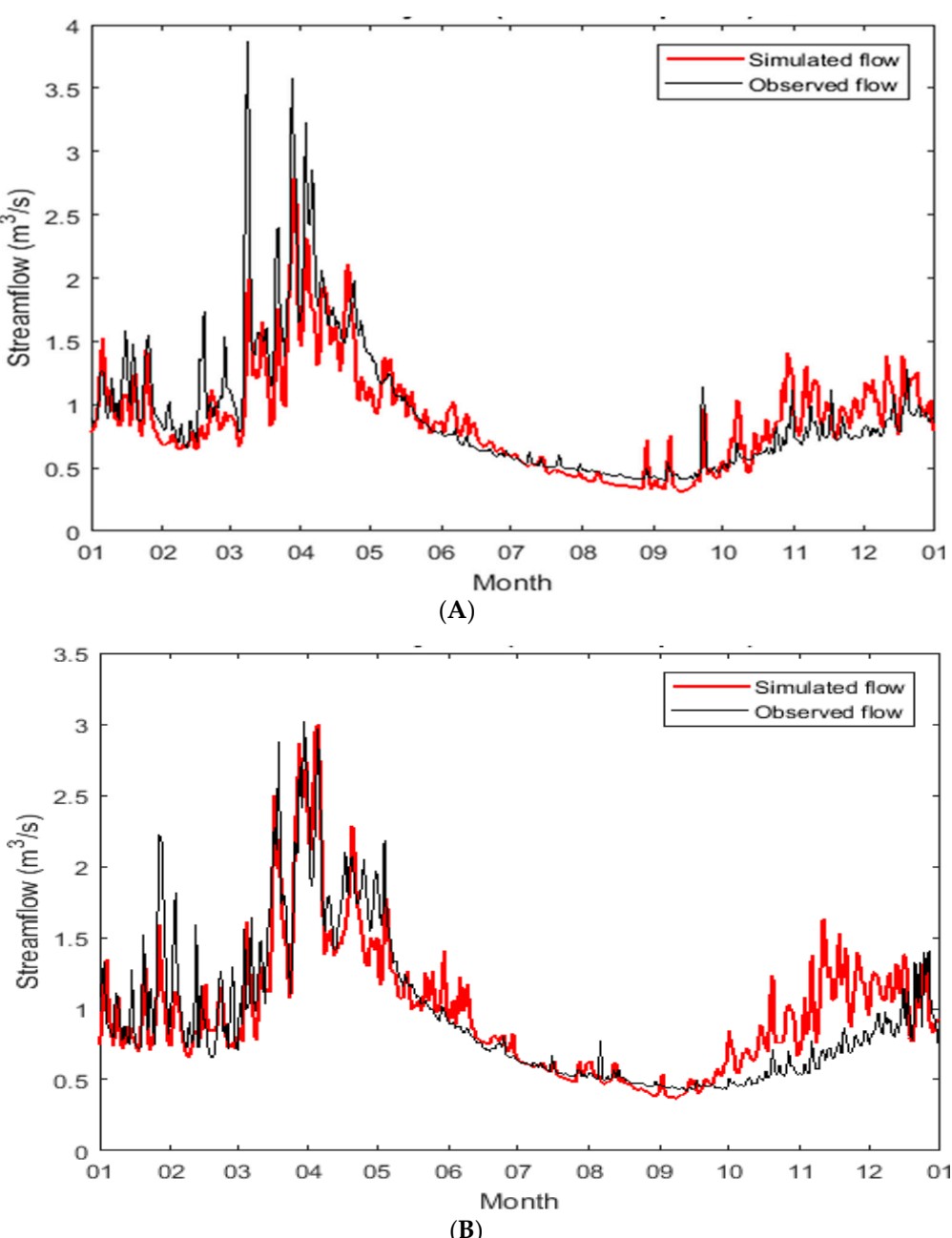

**Figure 4.** Daily interannual flow for the calibration period (**A**) and the validation period (**B**).

A sub-sample of simulations and input data are shown in Figure 5. It can be observed that some peak flows are overestimated by the model for this period. However, low flows, which are the focus of the present work because of their impact on water availability, water quality and aquatic habitat, appear to be relatively adequately simulated (e.g., months of January, February, August and September).

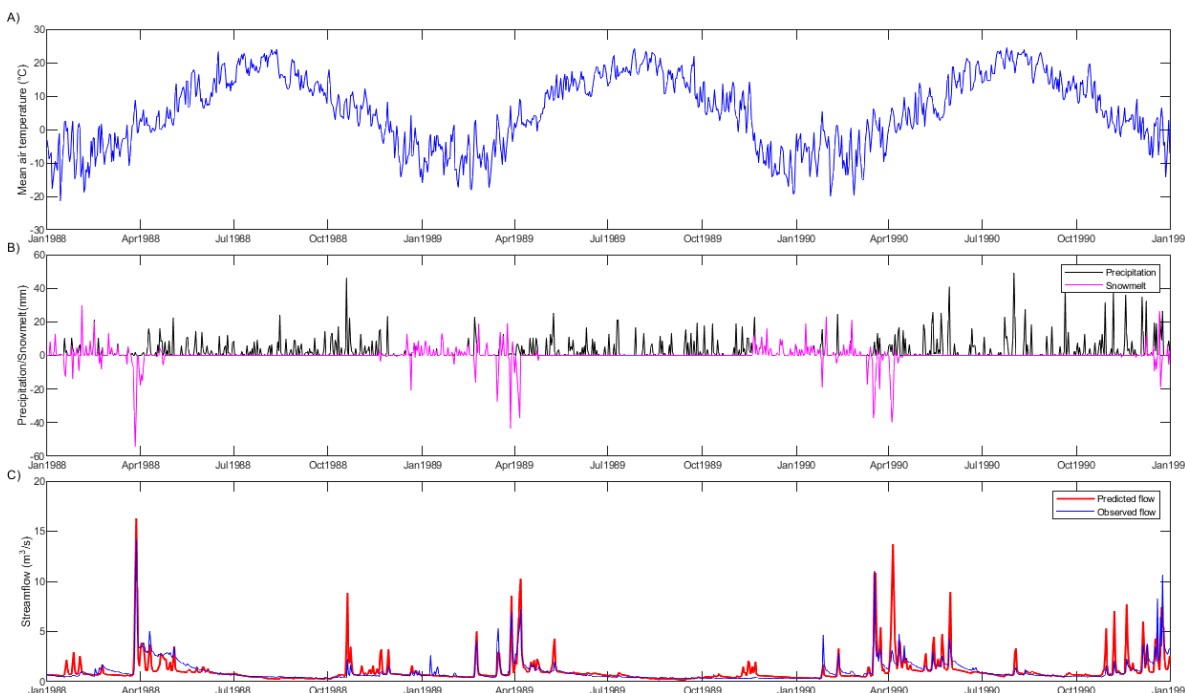

**Figure 5.** Observed mean air temperature (**A**), observed precipitation and simulated snowmelt over the contributing area (**B**), and simulated and observed flow at the hydrographic station for years 1988–1991 (**C**). Positive values for snowmelt indicate solid precipitation and negative values indicate snowpack melt.

### 3.3. Temperature Model Calibration and Validation

The seasonal pattern of water temperature was well reproduced by the temperature model (Figure 6). Maximum summer temperatures are overestimated for some years (2013 and 2016). During the winter, the model often underestimated water temperatures, with more frequent occurrences of 0 °C in the simulated temperatures than in the observation. Table 2 summarizes the performance statistics of the thermal calibration. Because of the strong seasonal signal in temperature time series, the KGE values for the temperature calibration and validation are much higher than for flows (75% < KGE < 95% by year). The optimal set of parameters (average values) yielded a KGE of 93.5% for the period 2013–2017 and a RMSE of 1.1 °C (Table 2).

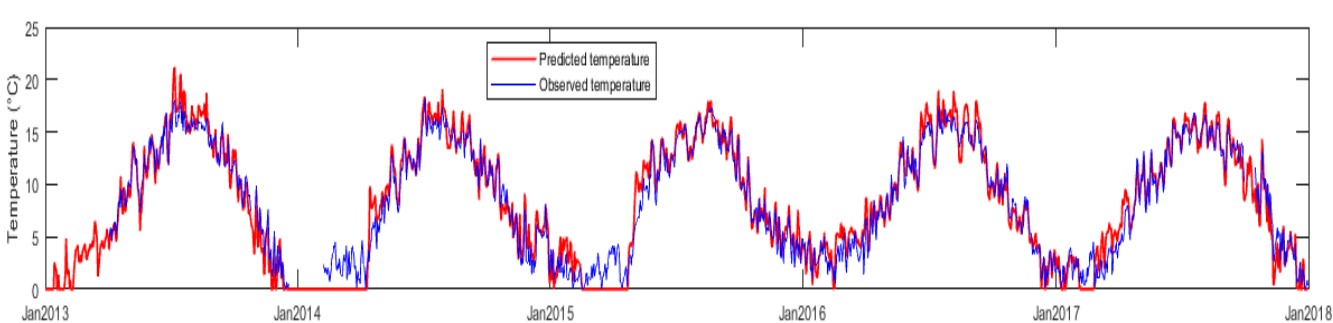

**Figure 6.** Observed and simulated water temperature at the temperature gauging station for years 2013–2018.

### 3.4. Climate Change Flow Scenarios

Median discharge during the low flow months of August and September is observed to increase compared to historical observations in all time periods and climate (Table 3). However, the extreme low flows ($Q_{95}$) are predicted to decrease between 45 to 63% depending on the scenarios/periods (Table 3). Therefore, according to the modelledled scenarios,

more fluctuations in the extreme low flows are to be expected, while the average (or median flows) will be trending upward.

**Table 2.** Performance criteria for the simulated water temperature using a 5-fold cross-validation method and performance criteria using the average of the parameters obtained for the cross-validation calibration samples.

|  | Validation Sample (5-Fold Cross-Validation) | | | | | Validation Sample (Average Parameters) |
|---|---|---|---|---|---|---|
|  | **2013** | **2014** | **2015** | **2016** | **2017** | **2013–2017** |
| KGE (%) | 79.66 | 86.56 | 93.00 | 89.33 | 95.16 | 93.54 |
| Bias (°C) | 0.00 | 0.71 | 0.40 | 0.11 | −0.34 | 0.20 |
| RMSE (°C) | 1.37 | 1.35 | 1.16 | 0.96 | 1.03 | 1.08 |

**Table 3.** Environmental flow statistics for August–September observations and two climate change scenarios on the Wilmot River.

|  | Observations | RCP 4.5 2018–2050 | RCP 4.5 2051–2100 | RCP 8.5 2018–2050 | RCP 8.5 2051–2100 |
|---|---|---|---|---|---|
| $Q_{50}$ (m³/s) | 0.45 | 0.49 | 0.59 | 0.57 | 0.48 |
| $Q_{95}$ (m³/s) | 0.33 | 0.17 | 0.16 | 0.18 | 0.12 |

Annual patterns can be observed from flow duration curves plotted for both climate change scenarios (Figure 7). When compared to observations, both scenarios predict a general increase in discharge except at the upper (>90% exceedance) tails of the exceedance curves, again reflecting more range of variability in low flow under the climate regimes examined. However, the lower tails of the flow duration curves (<10% exceedance) showed fewer differences suggesting extreme high flow events will be no more frequent than they are presently.

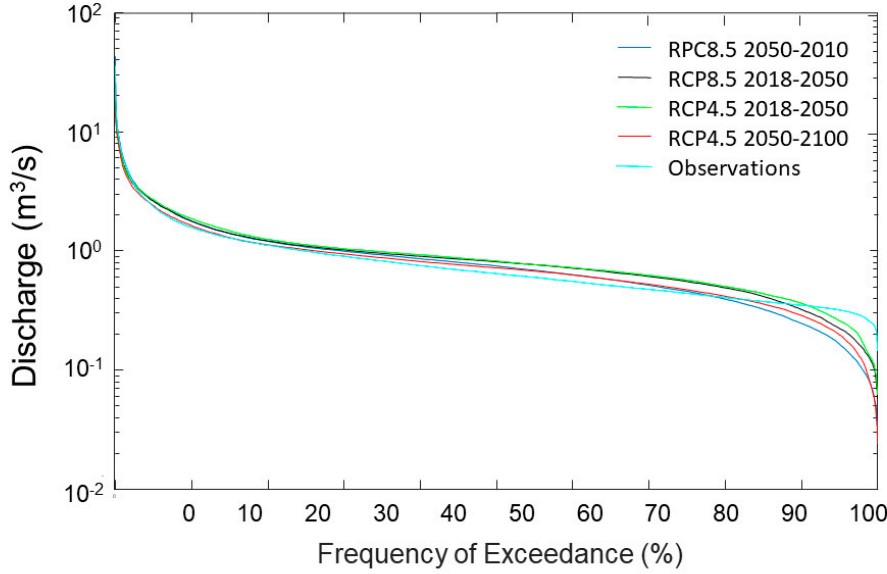

**Figure 7.** Flow duration curves for observed discharge, and discharge simulated using RCP 4.5 and RCP 8.5 scenarios.3.4. Climate Change Scenarios and Water Temperature.

When a linear trend is fitted to the time series of water temperature for scenario RCP 8.5 for 2018–2050 and 2051–2100, temperature increase over these periods is estimated at 0.88 °C and 1.8 °C, respectively. The Mann–Kendal test for trends was applied to all time series and these results are shown in Table 4. It can be seen that all increases are significant for the RCP 8.5 scenario, though the RCP 4.5 scenario shows few significant increases. The

more pessimistic (and more realistic) scenario shows water temperature trends between 0.25 and 0.4 °C/decade, depending on the month and the temperature statistic (minimum, mean, maximum; Table 4). According to this scenario, minimum water temperatures could frequently exceed 6 °C during low flow winter months (January–February; Figure 7). However, the change in summer low flow months (August–September) is less substantive than winter, with only approximately a 1 °C mean temperature changes under most scenarios. As with the flow data, variability is higher and peak summer temperature could exceed 20 °C, a threshold above which conditions become stressful for some fish species (e.g., brook trout), during the low flow years (Figure 8).

**Table 4.** Slope over time (°C/decades) for mean, minimum and maximum monthly simulated temperatures for Scenarios RCP 4.5 and RCP 8.5. Significant trends for temperature change are identified in bold, *p*-value in parentheses.

| Slope Parameter (°C/Decade) | January | February | August | September |
|---|---|---|---|---|
| **RCP 4.5** | | | | |
| Minimum | 0.009 (0.169) | 0.007 (0.085) | 0.008 (0.078) | -0.002 (0.904) |
| Mean | 0.005 (0.163) | 0.007 (0.068) | **0.011 (0.001)** | 0.0002 (0.055) |
| Maximum | 0.009 (0.074) | 0.007 (0.067) | 0.013 (0.020) | 0.002 (0.984) |
| **RCP 8.5** | | | | |
| Minimum | **0.25 (<0.001)** | **0.32 (<0.001)** | **0.29 (<0.001)** | **0.28 (<0.001)** |
| Mean | **0.24 (<0.001)** | **0.29 (<0.001)** | **0.35 (<0.001)** | **0.31 (<0.001)** |
| Maximum | **0.31 (<0.001)** | **0.29 (<0.001)** | **0.40 (<0.001)** | **0.32 (<0.001)** |

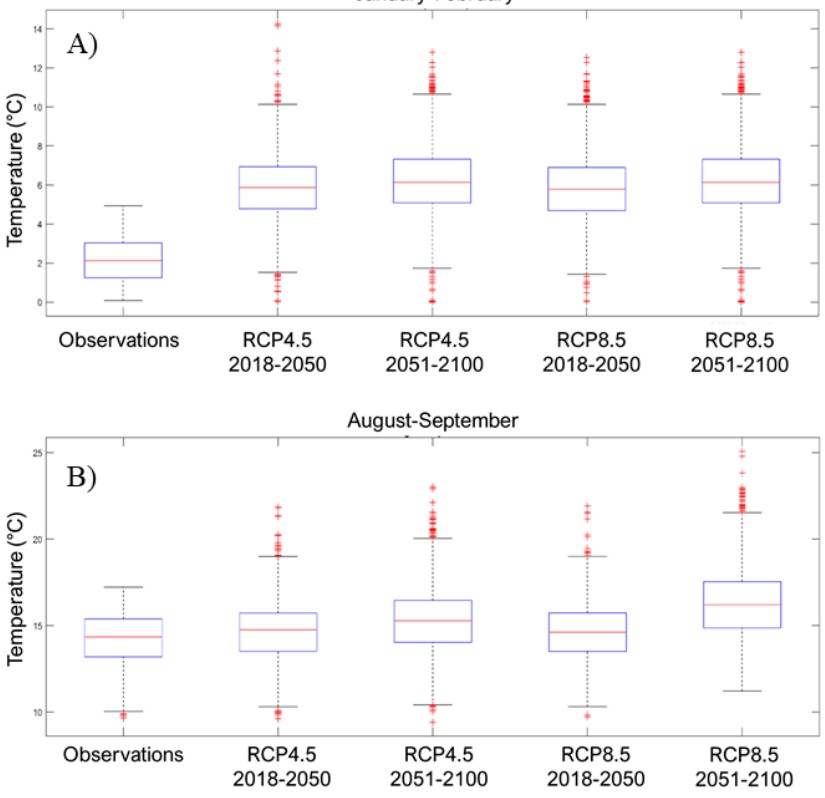

**Figure 8.** Box plots of water temperatures in low flow conditions for the months of (**A**) January–February and (**B**) August–September, for the two climate change scenarios and to future periods. For each box plot, the red line indicates the median, the bottom and top edges of the box indicate the 25th and 75th percentiles, respectively, the whiskers indicate the most extreme data points not considered outliers, and the outliers are indicated with the '+' red symbol.

## 4. Discussions and Conclusions

This first attempt at using the CEQUEAU deterministic hydrological and water temperature model to provide insight on possible future low flow and water temperature scenarios has shown that this tool has the potential to be implemented in PEI rivers. Our study showed that although median summer flow is not likely to change radically according to both climate change scenarios, more extreme low flows may occur on the Wilmot River. The model thermal scenarios indicate a likely significant increase in temperatures during the low flow periods.

The fractured sandstone aquifer of PEI dictates a groundwater-driven hydrological regime. This regime can be described by periods of groundwater recharge that start mid-fall and can continue into late spring, followed by a period of aquifer discharge through the summer months when evapotranspiration exceeds precipitation [27]. Groundwater base-flow has been estimated to be nearly equal to the stream flow in the summer months [28]. Annual groundwater recharge has been estimated to be 35–45% of annual precipitation [28]. Since much of the predicted temperature change in the region will be realized in winter, this could lead to less snow and more rainfall during that season, which in turn, will lead to less intense spring freshet. This may also lead to potentially higher groundwater recharge over the winter months. This is reflected in the model output herein as a generally higher median flow in the low flow months of August and September.

While the flow and temperature components of the model are predictive and based on historic data, the greatest uncertainly in predicting future flow are precipitation predictions derived from global climate models. Future work should apply those models to historic data to facilitate direct comparisons. It has been previously observed that the greatest discrepancies between regional climate precipitation models and observed historic precipitation occur in summer [28] potentially due to convective precipitation and several methods have been employed to address this persistent issue, including regional bias correction models [29]. Thus, climate regional downscaling remains a significant challenge, particularly with regards to precipitation projections.

Sources of natural climate variability may also be contributing to the uncertainly in precipitation projections. Global climate models as employed herein are well developed to describe external climate forcing. Internal variability refers to those random or chaotic processes that are not often captured by global climate models. Climate model internal variability can contribute 25–75% of total climate model uncertainty [30] and this natural variability may be irreducible. One significant source of such variability is the North Atlantic Oscillation (NAO), also called the Atlantic Multidecadal Oscillation. Multiple climate model ensembles have been successful at capturing some of this variability [31], even though unpredictable external forcing due to atmospheric aerosol from volcanic eruptions may drive some of this variability [32]. The single scenario models examined herein may not capture this natural variability as well as multi-model ensembles.

In contrast to precipitation, there is greater model certainty for regional temperature projections and hence the model predictions of future water temperature. Groundwater temperature on PEI reflects the mean annual surface air temperature and is consistently just over 7 °C [33] and strongly influences surface water temperature. With relatively short streams, there is not much opportunity for warming or cooling (days) before the water reaches the sea. In agreement with high-resolution climate model ensemble projections, the greatest temperature change in the region is projected to occur in winter, with only modest summer changes [34]. As predicted by the temperature model used herein, this would lead to greater warming of surface water in winter.

The first implication of the results presented herein is regarding ground/surface water extraction on PEI. Groundwater and surface extraction for agriculture remains a controversial issue in the region. Currently, surface water extraction in summer must cease when stream levels fall below 30% of the median flow (70Q50) for the period in question. Thus, future flow patterns will have direct implications on the potential to extract water sustainably. The examination of climate scenarios herein showed that while there may be

greater extreme low and high flow events, on average, there will be at least as much, if not more streamflow in the low flow months regardless of the climate scenario used.

A second implication of this work is the potential impacts of temperature change on native fishes. PEI supports two native coldwater fish species, brook trout (*Salvelinus fontinalis*) and Atlantic salmon (*Salmo salar*). Temperature regimes for those species are well established and the maximum weekly average temperature for growth is 19 °C and 20 °C for brook trout and Atlantic salmon, respectively [32]. Beyond this, significant stress occurs and the lethal temperature for short-term exposure of these species is 24 °C and 23 °C for brook trout and Atlantic salmon, respectively [32]. The models presented herein suggest that in the river studied, temperature increases will be less than 1 °C on average and will rarely reach 20 °C summer except in the 2051–2100 period for the most pessimistic scenario. Despite considerably higher winter temperature increases, temperature observations of rain events in winter show that snowmelt and runoff that causes streams to cool to 0 °C for short periods (authors unpublished data). These events are likely to be more frequent, with possible negative impacts on eggs survival.

Although the case study presented herein focused on one drainage basin, some of the conclusions may have repercussions on e-flow management in all PEI rivers. Future applications should allow us to not only to simulate future flow and water temperature scenarios using climate model outputs, but also to further investigate past trends by simulating past flow and temperature conditions to confirm the trends detected in the present study.

**Author Contributions:** C.C. produced all of the model analyses; he contributed to writing the paper; A.S.-H. completed some of the statistical analyses and contributed to writing the paper; M.R.v.d.H. contributed the water temperature data, completed some of the statistical analyses and contributed to writing the paper; T.B.M.J.O. validated the statistical analyses and contributed to writing the paper. All authors have read and agreed to the published version of the manuscript.

**Funding:** This research was funded in part by NSERC (grant 2019 06701) and the Canadian Water Network and in part by the Prince Edward Island Department of Environment, Energy, and Climate Action.

**Institutional Review Board Statement:** Not applicable.

**Informed Consent Statement:** Not applicable.

**Data Availability Statement:** Flow data were provided by Environment and Climate Change Canada. Temperature data are available in the RivTemp database (www.rivtemp.ca, accessed on 1 June 2018). The CEQUEAU model is available upon request to the corresponding author. An example of the input structure can also be made available upon request.

**Acknowledgments:** The authors wish to acknowledge the contribution of C. Calder for field work.

**Conflicts of Interest:** The authors declare no conflict of interest. 

## Nomenclature

| | | |
|---|---|---|
| *Q* | mm | Total runoff for a whole square |
| *P* | mm | Rain or snowmelt for a whole square |
| *ETP* | mm | Evapotranspiration for a whole square |
| *HU* | mm | Water accumulated in the upper reservoir for a whole square |
| *HL* | mm | Water accumulated in the lower reservoir for a whole square |

| | | |
|---|---|---|
| *CIN* | - | Percolation coefficient from the upper-zone to the lower-zone |
| *CVMAR* | | Lakes and marshes drainage coefficient |
| *CVNB* | | Lower-zone lower drainage coefficient |
| *CVNH* | | Lower-zone upper drainage coefficient |
| *CVSB* | | Upper-zone lower drainage coefficient |
| *CVSI* | - | Upper-zone intermediate drainage coefficient |
| *HINF* | mm | Percolation threshold from the upper to the lower-zone |
| *HINT* | mm | Upper-zone intermediate drainage threshold |
| *HM* | mm | Lakes and Marshes reservoir water level |
| *HMAR* | mm | Lakes and Marshes drainage threshold |
| *HN* | mm | Lower-zone reservoir water level |
| *HNAP* | mm | Lower-zone upper threshold |
| *HS* | mm | Upper-zone runoff reservoir water level |
| *HSOL* | mm | Upper-zone runoff threshold |
| *TRI* | % | Percentage of impervious area in the basin |
| *XKT* | - | Routing coefficient |
| *EXXKT* | | Routing coefficient fitting parameter |
| *RMA3* | $km^2$ | Area of the basin upstream of the partial square |
| *Sl* | $km^2$ | Area of the total surface water upstream of the partial square |
| *Slac* | $km^2$ | Area of surface water on the partial square |
| *CEKM2* | | Area of the whole square |
| $T_w$ | °C | Water temperature |
| *H* | MJ | Total enthalpy of the thermodynamic system |
| *V* | $m^3$ | Volume of water |
| *C* | $MJ/m^3/°C$ | Heat capacity of water (4.187) |

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
