# Peer review of "Water Temperature and Hydrological Modelling in the Context of Environmental Flows and Future Climate Change: Case Study of the Wilmot River (Canada)"

_water, doi:10.3390/w13152101_

Round 1
Reviewer 1 Report
The research was carried out using widespread approaches and methods. The conclusions of the work are clear and reasoned. I have only a few "technical" recommendations.
1. Keywords are required.
2. The scheme of Figure 1 is incomprehensible without the definition of abbreviations.
3. Figure 2. In this figure, the names of large water bodies and settlements (cities) are missing. They are necessary so that the international reader can better navigate the location of the investigated river basin.
4. The graph of precipitation/snowmelt in Figure 5. Why is the scale for snowmelt with minus values?
5. Figure 6. Is this the temperature at the surface of river water? Sub-ice water temperatures are above zero (except when the river completely freezes to its bottom).
6. Table 2. Average parameters for 2013-2017 are calculated incorrectly. Please recalculate them again.
7. Table 3. What are the units of measurement?
Author Response
Answers to Reviewer #1 comments:
The authors are grateful to the reviewer for his/her comments which contributed to the improvement of the quality of the paper. We provide hereafter the answers to the reviewer’s comments.
Comment 1: Keywords are required.
Answer: Keywords were added in the revised version of the manuscript.
Comment 2: The scheme of Figure 1 is incomprehensible without the definition of abbreviations.
Answer: A nomenclature with all symbols used in the paper was added at the end of the manuscript.
Comment 3: Figure 2. In this figure, the names of large water bodies and settlements (cities) are missing. They are necessary so that the international reader can better navigate the location of the investigated river basin.
Answer: The only major city, Summerside PEI has been added as have the larger bays, with the larger bodies they flow into indicated in parentheses.
Comment 4: The graph of precipitation/snowmelt in Figure 5. Why is the scale for snowmelt with minus values?
Answer: In Figure 5, positive values for the snowmelt curve represent snow accumulation on the ground while negative values represent snowmelt. This is explained in the revised version of the manuscript in the caption of Figure 5.
Comment 5: Figure 6. Is this the temperature at the surface of river water? Sub-ice water temperatures are above zero (except when the river completely freezes to its bottom).
Answer: Figure 6 compares the river temperature simulated by the model against observations. PEI rivers are typically well-mixed, except in deep pools (which were not monitored in the present study). Also, the relatively large contribution of ground water in PEI rivers is such that virtually no rivers freeze to the bottom in the winter, as shown in the observations (blue line). Our model was not fully able to represent this phenomenon consistently. Winter simulated temperatures were often at 0°C, while observations only reached the freezing point episodically. This is now recognized in the manuscript in Section 3.2.
Comment 6: Table 2. Average parameters for 2013-2017 are calculated incorrectly. Please recalculate them again.
Answer: The column “Validation sample (Average parameters)” represents the validation statistics obtained with the mean value of the parameters for each sub-sample model. It does not represent the averages of the performance statistics. This information was provided in line 176 of the original manuscript: “The average value of each parameter is used for the final thermal simulation.”
Comment 7: Table 3. What are the units of measurement?
Answer: Units were added in Table 3.
Reviewer 2 Report
The evaluated manuscript is entitled “Hydrological and Water Temperature Modelling in the Context of Environmental Flows and Climate Change: case study of the Wilmot River (Canada)”. The text is written by Charron and co-authors from Canada. It presents the modeling of the climate change impact on the precipitation transformation into the outflow. The applied software is called CEQUEAU. In general, it’s a model simulating hydrological and water temperature transformations. The approach presented differs from typical rainfall-runoff modeling, because snowpack and melting are also taken into account. The text is nicely written and the main concepts are well presented. However, there are also some problems. The concerns are described below in three groups: Main problems, Detailed remarks, and Minor suggestions.
Main problems:
- Definition of the purpose
The purpose is well defined in lines 78-79.
“The objective of the present study is to test the implementation of the CEQUEAU hydrological and water temperature model [17, 18] to simulate both flow and water temperatures on one PEI drainage basin: the Wilmot River.”
Although it is understandable, we have to notice that such a purpose is not scientific. I suggest reformulating this sentence, to indicate the scientific values of the possible results.
- Description of the model
This part of the text is not satisfactory. The applied model CEQUEAU was implemented and described previously. Hence, the authors could think that the detailed description of the model elements is not necessary. In my opinion, isn’t the correct approach. First of all, some of the previous papers are written in French (e.g. [17]). It’s a very popular and beautiful language, but not so spread as English. So it may be a problem to follow ideas presented there for some readers of the Water journal. Others are presented during local conferences (e.g. [18]) and their availability may be problematic. Secondly, the elements of the model are important for the proper understanding of the applied concept. These are also crucial for the interpretation of the results. So, the description of the model has to be improved. There are more details in the next section.
- Description of the model calibration and verification.
This part of the description is incomplete. As it’s indicated in the detailed remarks it’s not known exactly what kind of method is applied, what is the objective function, what is the relationship between the method, objective function, and additional criteria applied in the verification. Some of the descriptions are even confusing.
Detailed remarks
- Subsection 2.1 Trend analysis
Supposedly this is done to detect the trends in the precipitation and interpret them as an effect of climate change. The methods described here should be better explained. And the relation to the climate change scenarios RCP4.5 and RCP8.5 should also be indicated.
- Lines 101-103
How the so-called “whole squares” are determined? Do you use any GIS tools to make such decomposition? It looks like computations over the raster cells. If there is some difference, it could be good to indicate it.
- Lines 104-105
How the “whole squares” are partitioned into “partial squares”? Some figures may be useful to explain this idea. It's similar to the hybrid mesh concept with one mesh inside another. But there might be some references to proper bibliography.
- Line 107
How the downstream direction for a particular cell is determined? Is the local slope calculated? How?
- Line 110
What do you mean by "advection"? It could be interpreted in several ways. Do you mean hypothetical streamflow from one cell to another?
- Lines 110-111
Is the concept of “interconnected reservoirs” related to typical hydrological models created as cascades of linear/nonlinear reservoirs? Such concepts were applied in hydrology and these are still very useful. In the Figure, there is no "series". There are only three reservoirs, each representing a different process.
- Line 114
"Liquid precipitation"? Is it simply rainfall? Do you mean something else?
- Line 122
How is XKTi used? What is the role of this coefficient in the model?
- Line 123
The “area of the basin upstream” is calculated as the total area or only the area inside the partial square? Or maybe it's an area upstream in the whole square?
- Line 123-124
Is the coefficient Sl the same as the area of inundation?
- Line 124
“surficial water” = surface water?
- Lines 137-138
“a split sample method” – What does it mean? The sample was split? And what next? One part used for identification, another applied for verification? It's not a specific method, but an obvious approach.
- Lines 140-142
If there is one-dimensional optimization, there should be a single objective function. If there are several objective functions, we have Pareto multi-dimensional optimization and basic search is not enough.
Supposedly optimization is one-dimensional, a basic search method is applied to solve the problem with the scalar objective function, and then the results are verified using additional criteria. Is it the concept behind the presented approach?
- Line 146
"observed value" of what... water stage, discharge, temperature, ...
- Line 153
How is it determined the percentage of land cover? Have you used some databases on land use/cover?
- Line 154
“whole squares 0.25 km2 each” – It means 0.5 km x 0.5 km. The resolution is rather coarse. It's the weakness of this model. An approach based on split into watersheds and sub-watersheds should be more effective. Could you comment?
- Lines 159-163
In my opinion, this passage should be in the section where the model is described and the optimization is mentioned the first time.
- Line 169
“a resolution of 0.33° x 0.33°” – This resolution could be expressed in the same units as the resolution of the entire hydrological model.
- Line 176
"repeated for each year", but "calibrated on four-year sub-sample"... What is what? It should be explained better.
- Lines 194-195
“approximately 40 x 40 km” – It's very coarse. To apply such data some kind of downscaling is necessary. Could you comment?
- Lines 195-197
Are the mentioned models applied by the authors of the paper, or these are public globally available results.
- Line 197
"grid point closest” ? No interpolation?
- Line 235 and Table 1
The given values and description suggest that the analyzed measures were calculated after the optimization process. In such a case none of these is the objective function. The objective function and the optimization process are not described properly.
I guess that the table doesn’t show the comparison of calibration and verification. It rather presents two stages of verification, one done with dependent data and the second done with independent data. But calibration is something that happened before.
Minor suggestions
- Line 26
Lack of keywords
- Line 158
Figure 2 ?
- Line 225-226
“To confirm this and ANOVA […]” – grammar problem
- Figure 3
In the caption of the figure, there could be given some information about the lines which are shown. How were they created?
- Section 4 Discussion and conclusions
In my opinion, this section is rather long. Please, consider the split of this part into separate sections discussion, and conclusions.
Author Response
The authors are grateful to the reviewer for his/her comments which contributed to the improvement of the quality of the paper. We provide hereafter the answers to the reviewer’s comments.
Main problems:
Comment 1: Definition of the purpose
The purpose is defined in lines 78-79.
“The objective of the present study is to test the implementation of the CEQUEAU hydrological and water temperature model [17, 18] to simulate both flow and water temperatures on one PEI drainage basin: the Wilmot River.”
Although it is understandable, we have to notice that such a purpose is not scientific. I suggest reformulating this sentence, to indicate the scientific values of the possible results.
Answer:
The objective statement was re-written and now reads:
The objective of the present study is to investigate current and possible future hydrological and thermal conditions during low flow months on a small agricultural watershed in the context of re-visiting environmental flows guidelines. This is done by implementing the CEQUEAU hydrological and water temperature model [17, 18] to simulate both flow and water temperatures on one PEI drainage basin: the Wilmot River.
Comment 2: Description of the model
This part of the text is not satisfactory. The applied model CEQUEAU was implemented and described previously. Hence, the authors could think that the detailed description of the model elements is not necessary. In my opinion, isn’t the correct approach. First of all, some of the previous papers are written in French (e.g. [17]). It’s a very popular and beautiful language, but not so spread as English. So it may be a problem to follow ideas presented there for some readers of the Water journal. Others are presented during local conferences (e.g. [18]) and their availability may be problematic. Secondly, the elements of the model are important for the proper understanding of the applied concept. These are also crucial for the interpretation of the results. So, the description of the model has to be improved. There are more details in the next section.
Answer:
Reference [18] is readily available on line. Reference [23], which was in French was replaced by a reference in English. Additional information is now provided on the main components of the hydrological budget and those of the heat budget.
Comment 3: Description of the model calibration and verification.
This part of the description is incomplete. As it’s indicated in the detailed remarks it’s not known exactly what kind of method is applied, what is the objective function, what is the relationship between the method, objective function, and additional criteria applied in the verification. Some of the descriptions are even confusing.
Answer:
In the revised version, the calibration/validation methods (split sample for the hydrological module and jackknife cross-validation for the thermal module are specified. The objective function (KGE) is also specified. Additional criteria are presented in Equations 4 and 5.
Detailed remarks
Comment 4: Subsection 2.1 Trend analysis
Supposedly this is done to detect the trends in the precipitation and interpret them as an effect of climate change. The methods described here should be better explained. And the relation to the climate change scenarios RCP4.5 and RCP8.5 should also be indicated.
Answer: The paragraph was modified: “Trend analyses were performed on historical precipitation time series and on future precipitation scenarios (described hereafter) in order to determine if either RCP4.5 and/or RCP8.5 climate change scenario projected a statistically significant change in precipitation as precipitation is a major component driving stream flow.” This now explains the purpose of the analysis and the relation to the climate change scenarios.
Comment 5: Lines 101-103
How the so-called “whole squares” are determined? Do you use any GIS tools to make such decomposition? It looks like computations over the raster cells. If there is some difference, it could be good to indicate it.
Answer:
Whole squares are determined by imposing a grid on the drainage basin. The size of the squares is determined subjectively as a trade-off between having a sufficiently fine resolution to define the main watershed features (e.g. tributaries, lakes, etc.) and coarse enough to save computing time.
Comment 6: Lines 104-105
How the “whole squares” are partitioned into “partial squares”? Some figures may be useful to explain this idea. It's similar to the hybrid mesh concept with one mesh inside another. But there might be some references to proper bibliography.
Answer:
Additional information was provided in Section 2.2. It now reads:
“For each whole square altitude, percentage of forest area and percentage of area covered by lakes and wetlands must be defined. Water routing is defined initially by partitioning whole squares into a maximum of four so-called “partial square” according to the water divides. This subdivision of up to four partial squares in each whole square allows to define water routing at the proper scale.
Comment 7: Line 107
How the downstream direction for a particular cell is determined? Is the local slope calculated? How?
Answer:
Yes, the downstream direction is based on slopes between partial squares.
Comment 8: Line 110
What do you mean by "advection"? It could be interpreted in several ways. Do you mean hypothetical streamflow from one cell to another?
Answer:
Advection is commonly defined as “the transport of a substance or quantity by bulk motion of a fluid.” Therefore, heat advected is the quantity of heat associated with water movement (i.e. flow).
Comment 9: Lines 110-111
Is the concept of “interconnected reservoirs” related to typical hydrological models created as cascades of linear/nonlinear reservoirs? Such concepts were applied in hydrology and these are still very useful. In the Figure, there is no "series". There are only three reservoirs, each representing a different process.
Answer:
As shown in Figure 1, two reservoirs (Upper and Lower soil zones), are connected (white arrow showing possible infiltration). The reservoir representing lakes and marshes is not connected to the soil zones, Lakes and marshes contribute to runoff when they “spill” above the HMAR threshold (see Figure 1). The text in Section 2.2 was modified and provides more details.
Comment 10: Line 114
"Liquid precipitation"? Is it simply rainfall? Do you mean something else?
Answer: “Liquid precipitation” means rainfall. The wording was changed.
Comment 11: Line 122
How is XKTi used? What is the role of this coefficient in the model?
Answer:
XKTI is a routing coefficient. It is a fraction (0< XKTI<1) based on the ratio of partial square areas to upstream surface area. It is assigned as a weighting coefficient on the runoff from upstream to determine how much of the upstream runoff is attributed to each partial square downstream at each time step. Description of its calculation is provided in Equation (2) and in the text immediately below.
Comment 12: Line 123
The “area of the basin upstream” is calculated as the total area or only the area inside the partial square? Or maybe it's an area upstream in the whole square?
Answer:
It is the total area and this is now specified in the revised manuscript.
Comment 13: Line 123-124
Is the coefficient Sl the same as the area of inundation?
Answer:
We are not certain how the reviewer defines “area of inundation”, so we cannot answer this question.
Comment 14: Line 124
“surficial water” = surface water?
Answer: surficial water is the surface water. This is corrected in the revised version of the manuscript for clarity.
Comment 15: Lines 137-138
“a split sample method” – What does it mean? The sample was split? And what next? One part used for identification, another applied for verification? It's not a specific method, but an obvious approach.
Answer: The authors agree with the reviewer that the explanation was incomplete. The method is more clearly explained in the revised version of the manuscript.
Comment 16: Lines 140-142
If there is one-dimensional optimization, there should be a single objective function. If there are several objective functions, we have Pareto multi-dimensional optimization and basic search is not enough.
Supposedly optimization is one-dimensional, a basic search method is applied to solve the problem with the scalar objective function, and then the results are verified using additional criteria. Is it the concept behind the presented approach?
Answer: Only one metric was used for the optimization of the parameters. The RMSE and Bias were used as performance metrics but not for model parameter optimization. This is clearly explained in the revised version of the manuscript.
Comment 17: Line 146
"observed value" of what... water stage, discharge, temperature, ...
Answer: the observed value means either the water discharge or the temperature. This section of the text was written in a general manner to avoid repetition. However, flow and temperature are now mentioned initially.
Comment 18: Line 153
How is it determined the percentage of land cover? Have you used some databases on land use/cover?
Answer:
Yes, databases on land use/land cover were used.
Comment 19: Line 154
“whole squares 0.25 km2 each” – It means 0.5 km x 0.5 km. The resolution is rather coarse. It's the weakness of this model. An approach based on split into watersheds and sub-watersheds should be more effective. Could you comment?
Answer:
As stated earlier, the resolution is selected subjectively, based on a trade-off between detailed watershed characterization and computing time. In this study, temperature data were scarce and only available at one station. Using a finer resolution appeared to be a moot point. A study by Dugdale et al. (2017) showed that “despite some model strength fluctuations linked to variability in computed basin size/land-use, only a minor decrease in model strength (mean NSE reduction = 0.03) was observed at relatively fine resolutions”.
Comment 20: Lines 159-163
In my opinion, this passage should be in the section where the model is described and the optimization is mentioned the first time.
Answer: The authors agree with the reviewer that this text belongs to the section 2.2. It was thus moved and reformulated for integration in the right section.
Comment 21: Line 169
“a resolution of 0.33° x 0.33°” – This resolution could be expressed in the same units as the resolution of the entire hydrological model.
Answer:
Climate model and reanalyses resolutions are always provide in degrees because one degree of latitude varies in size according to its location on the planet.
Comment 22: Line 176
"repeated for each year", but "calibrated on four-year sub-sample"... What is what? It should be explained better.
Answer:
A jackknife procedure excludes one year and uses all of the others for calibration. This procedure is repeated for all years.
Comment 23: Lines 194-195
“approximately 40 x 40 km” – It's very coarse. To apply such data some kind of downscaling is necessary. Could you comment?
Answer:
The reviewer is right to highlight this. However, this is the result of a dynamic downscaling exercise. We chose not to further downscale because our objective is to compare past and future conditions at the same scale. Although this may impart some errors and biases, the approach was used to allow for relative comparison and for consistency.
Comment 24: Lines 195-197
Are the mentioned models applied by the authors of the paper, or these are public globally available results.
Answer:
CEQUEAU is fully available upon request.
Comment 25: Line 197
"grid point closest”? No interpolation?
Answer: Resolution is rather coarse for the size of the basin under study. Interpolation will not produce results significantly different from the nearest point.
Comment 26: Line 235 and Table 1
The given values and description suggest that the analyzed measures were calculated after the optimization process. In such a case none of these is the objective function. The objective function and the optimization process are not described properly.
I guess that the table doesn’t show the comparison of calibration and verification. It rather presents two stages of verification, one done with dependent data and the second done with independent data. But calibration is something that happened before.
Answer:
Tables 1 and 2 provide the KGE value, which is the objective function, for calibration and validation. Validation performance was always computed on data that were not used for calibration. Hence they are independent.
Minor suggestions
Comment 27: Line 26
Lack of keywords
Answer: Keywords were added in the revised manuscript.
Comment 28: Line 158
Figure 2?
Answer: The authors mean Figure 2 as suggested by the reviewer. The mistake was corrected. The authors thank the reviewer for pointing out this blunder.
Comment 29: Line 225-226
“To confirm this and ANOVA […]” – grammar problem
Answer: This grammar was corrected.
Comment 30: Figure 3
In the caption of the figure, there could be given some information about the lines which are shown. How were they created?
Answer: The solid line represents the linear regression while the dashed lines represent the confidence intervals. This is now specified in the revised version of the manuscript.
Comment 31: Section 4 Discussion and conclusions
In my opinion, this section is rather long. Please, consider the split of this part into separate sections discussion, and conclusions.
Answer:
Although we decided to keep the sections together, some of the text was deleted to shorten it.
Reviewer 3 Report
Charron et al.
Hydrological and Water Temperature Modelling in the Context of Environmental Flows and Climate Change: case study of the Wilmot River (Canada)
Brief summary
Anthropogenic climate change is expected to affect environmental stream flows and temperature with consequences for cold-water fishes and water abstraction. A hydrological and water temperature numerical model is combined with two greenhouse gas scenarios and applied on the Wilmot River, Canada. The hydrological module divides the watershed into hydrological units characterized by forest cover, lakes, and wetlands. Vertical and lateral water flow and temperature are calculated depending on meteorological data and flow resistance. Simulations predict from stable to increased streamflow and increased variability. Water temperature is expected to increase from 1°C - 4°C with Implications for cold-water fish and water abstraction. The changes are judged to be not severe except for periodic extremes.
Broad comments
The paper is important for resource decision makers planning for future impacts of anthropogenic climate change. The workflow presented potentially inspires similar environmental studies.
The abstract should highlight the importance of anthropogenic climate change on environmental flow (e-flow) in the beginning of the abstract.
All variables in equations, figures, and text must be defined and their units given.
The description of the CEQUEAU model in English is valuable but it needs clarification. Addition of a figure illustrating the lateral flow is needed. The figure illustrating the vertical flow is hard to understand. I suggest drawing and coloring the various reservoirs to mimic the natural reservoirs. It should be clear where the water is.
Input data for CEQUEAU should be stored in a repository, at least one example.
Specific comments
- 2. Please change the tittle: be more precise on the meaning of “Hydrological temperature”, “environmental flows”, “climate change future or past” be more precise.
- 10. water flow: specify where it is, surface or groundwater?.
- 12. No published studies…. : Move sentence to the introduction.
- 14-15. Remove (). Move information on year intervals to the introduction. Can conclusions in connection with the Wilmot River be generalized to other areas?
- 15-17. Move these important sentences about future climate change to after the first sentence. Highlight why this is important to decision makers. Make it clear to the reader that the article is about the anthropogenic climate change, not so much natural change.
- 16. most pessimistic: with regards to climate change?
- 19. was projected: do you mean is expected?
- 21. (e.g. 95th percentile): Move this information to the introduction.
- 96. Bray-Curtis dissimilarity: What is it?
- 99. semi-distributed: What is the meaning?
- 101. equal surfaces: surfaces of equal area?
- 102. Forest area?
- 104. Please help the reader by adding a figure showing “whole square” and “partial squares” and arrows indicating routing and receiving partial square.
- 106. Please remind the reader the meaning of “interflow and base flow”.
- 108. production function: Consider changing the wording. Water is not produced but transported. I suggest: CEQUEAU is composed of two main functions, the vertical flow function responsible for quantifying the vertical flow of water …. Please consider changing this wording also other places in the document.
- 111. ground: Please define the ground, how much does it include?
- 113. Please insert also () around the HL’s. Rewrite this equation and be precise on the meaning of t. Is it the time or is it the time step? I suggest: Q(tn) = P(tn) - …….HU(tn-1) ……., where tn is the time at time step n for n = 1,...,N, where N is the total number of time steps. Please inform the reader: how large is a time step, days, months?
- 114. The unit is here given as mm. Is per year, month? Please explain how “millimeter” is related to volume per time for all variables. Is the meaning total runoff per whole square.
- 116. t is the model time step: in l. 112 it is defined to be the time. I think you mean it is the present time step and t-1 is the preceding time step, and that t = 1,.. N, where N is the number of time steps. Please be clear with the notation.
- 118. Please repeat that “production function” means vertical flow. Add Q(tn) both in the figure caption and in the figure. Explain all abbreviations and give units of variables in the figure. Explain also what is “Upper and lower zone”? The figure needs much clarification.
- 119. Equation 2: XKT i is a dimensionless quantity according with the equation. In line 119 it is called the transfer function while in line 122 it is called the routing coefficient. Please provide a clear definition. Subscripts i are missing for variables on the right side of the equation.
- 123. basin: what is the meaning. Is it a whole square or partial square?
- 126. hydrological simulations: hydrological simulation results.
- 129. Temperature: Change of temperature.
- 130-136. Tw is not the temperature but the change of temperature since H is also changing with time. Please give consistent units to all variables. Please add a dot after EXXKT. And replace * with a dot.
- 137-138. Split sample method: Please explain.
- 164-170. Please move these explanations to the first reference to fig. 2.
- 179. Figure 2: The red box on the Canada box is rotated relatively to the large map. CORDEX and NARR must be explained in connection with first reference to fig. 2.
- 189. GHG: (Green House Gas).
- 189. RCP 4.5: RCP4.5.
- 196-197. Please give references.
- 203. Please indicate the calibration and the evaluation periods in fig. 3.
- 209-210. Not shown in fig. 3. Where is it shown?
- 214. Figure 2B: Figure 3B.
- 216-218. Please reformulate this sentence. It is hard to understand especially the meaning of the last part. And what do you mean by patterns.
- 218. climate change scenarios: Do you mean the time intervals given in the previous line?
- 226. This sentence is hard to understand, what is ANOVA?
- 246. low flows, which are the focus of the present work: Please explain why this is the focus. Why is it important? And since it says here that it is the focus it should be mentioned much earlier in the paper eventually in the abstract.
- 230. Please indicate calibration and validation periods in the figure.
- 255. Note the three figures with A, B, C and also in the figure text.
- 263. final set of parameters: What is the meaning of final?
- 274-275. Where is the table?
- 275. decrease by 45-63%.: Please indicate when.
- 288. Please specify period 1 and 2.
- 288. Flow duration: Flow exceedance probability.
- 289. Temperature: change of temperature. Check throughout the paper and replace “temperature” to “change of temperature”.
- 291. Remove the last RCP8.5, not necessary to repeat it.
- 304. Change of temperature, not temperature! And add the unit of temperature.
- 304. Slope over time: Please add the unit ◦C / month, ◦C / year ?.
- 304. Month January February August September: Remove this line a little down in the table. It is clear from the table column head.
- 305. Significant trends: Significant trends for temperature change, ◦C.
- 306. Please explain the meaning of blue boxes, read and black lines.
- 307. Please split discussion and conclusion into two separate sections. The conclusion section should be short with a few important results.
- Please repeat the meaning of PEI.
- 320. 100% of: equal to.
- 376. Potential impact: The direct temperature impact on fish are discussed. What about indirect effects such as on nutrients?
- 465. Please correct the reference [23] GUY MORIN , DENIS COUILLARD , DANIEL CLUIS , H. GERALD JONES & JEAN-MAURICE GAUTHIER (1987) Prévision des températures de l'eau en rivière à l'aide d'un modèle conceptual, Hydrological Sciences Journal, 32:1, 31-41, DOI: 10.1080/02626668709491160

Author Response
The authors are grateful to the reviewer for his/her comments which contributed to the improvement of the quality of the paper. We provide hereafter the answers to the reviewer’s comments.
Broad comments
Comment 1: The abstract should highlight the importance of anthropogenic climate change on environmental flow (e-flow) in the beginning of the abstract.
Answer:
The first sentence of the abstract was modified to provide this context.
Comment 2: All variables in equations, figures, and text must be defined and their units given.
Answer: A nomenclature with all the symbols used in the paper was added at the end of the manuscript.
Comment 3: The description of the CEQUEAU model in English is valuable but it needs clarification. Addition of a figure illustrating the lateral flow is needed. The figure illustrating the vertical flow is hard to understand. I suggest drawing and coloring the various reservoirs to mimic the natural reservoirs. It should be clear where the water is.
Answer:
As stated before, we provided clarification in the text. The upstream-downstream flow routing is scheme is also explained in greater details. We believe that an additional figure would lengthen the paper unduly, as it already has 8 figures.
Comment 4: Input data for CEQUEAU should be stored in a repository, at least one example.
Answer:
It is now stated in the Data Availability Statement that the model is available upon request and an example of the input data structure can also be provided.
Specific comments
Comment 5:
- Please change the tittle: be more precise on the meaning of “Hydrological temperature”, “environmental flows”, “climate change future or past” be more precise.
Answer: the title was modified and now reads: “Water Temperature and Hydrological Modelling in the Context of Environmental Flows and Future Climate Change: case study of the Wilmot River (Canada)”
The term “hydrological temperature” does not appear in the text. The term Environmental flow is defined in the first paragraph of the introduction. The words “climate change future or past” do not appear together in the text.
Comment 6:
- water flow: specify where it is, surface or groundwater?.
Answer: In the revised version of the manuscript it is specified that we are dealing with “surface water flow”.
Comment 7:
- No published studies…. : Move sentence to the introduction.
Answer: This sentence was moved to the introduction.
Comment 8:
14-15. Remove (). Move information on year intervals to the introduction. Can conclusions in connection with the Wilmot River be generalized to other areas?
Answer: The parenthesis were removed and information on year intervals was moved to the introduction. We now state in the conclusion that “Although the case study presented herein focused on one river, some of the conclusions may have repercussions on e-flow management in all PEI rivers.”
Comment 9:
15-17. Move these important sentences about future climate change to after the first sentence. Highlight why this is important to decision makers. Make it clear to the reader that the article is about the anthropogenic climate change, not so much natural change.
Answer: These sentences were moved after the first sentence. And the first sentence now reads: “Simulation of surface water flow and temperature under a non-stationary, anthropogenically impacted climate is critical to decision makers….”
Comment 10:
- most pessimistic: with regards to climate change?
Answer: Yes. This is clearly indicated in the revised version of the manuscript.
Comment 11:
- was projected: do you mean is expected?
Answer: “expected” is a better term than “projected”. The change is made in the revised version of the manuscript. The authors thank the reviewer for the suggestion.
Comment 12:
- (e.g. 95th percentile): Move this information to the introduction.
Answer:
The text in parentheses was deleted.
Comment 13:
- Bray-Curtis dissimilarity: What is it?
Answer:
Additional information is provided in Section 2.1. Note that there was an error here, the Bray-Curtis similarity index was used.
Comment 14:
- semi-distributed: What is the meaning?
Answer:
A semi-distributed model is a hybrid between a lumped model (i.e. all parameters have a single value for the entire drainage basin) and a fully distributed model (i.e. all parameters vary spatially).
Comment 15:
- equal surfaces: surfaces of equal area?
Answer: The text was corrected.
Comment 16:
- Forest area?
Answer: “Forest cover” was replaced by “forest area”.
Comment 17:
- Please help the reader by adding a figure showing “whole square” and “partial squares” and arrows indicating routing and receiving partial square.
Answer:
We decided against adding more figures, as their number is already high. However, as stated before, the description of partial squares was improved.
Comment 18:
- Please remind the reader the meaning of “interflow and base flow”.
Answer:
Definitions of interflow and baseflow are now provided.
Comment 19:
- production function: Consider changing the wording. Water is not produced but transported. I suggest: CEQUEAU is composed of two main functions, the vertical flow function responsible for quantifying the vertical flow of water …. Please consider changing this wording also other places in the document.
Answer:
We cannot change the wording because all of the papers previously published on the CEQUEAU model use this terminology.
Comment 20:
- ground: Please define the ground, how much does it include?
Answer:
The sentence was changed to: “The production function is modeled by a series of reservoirs, two of which are connected, as they estimate the soil infiltration and soil water storage.”
Comment 21:
- Please insert also () around the HL’s. Rewrite this equation and be precise on the meaning of t. Is it the time or is it the time step? I suggest: Q(tn)= P(tn) - …….HU(tn-1) ……., where tnis the time at time step n for n = 1,...,N, where N is the total number of time steps. Please inform the reader: how large is a time step, days, months?
Answer:
The Equation was changed and the daily time step specified.
Comment 22:
- The unit is here given as mm. Is per year, month? Please explain how “millimeter” is related to volume per time for all variables. Is the meaning total runoff per whole square?
Answer:
Units are mm at the daily time step.
Comment 23:
- t is the model time step: in l. 112 it is defined to be the time. I think you mean it is the present time step and t-1 is the preceding time step, and that t = 1,.. N, where N is the number of time steps. Please be clear with the notation.
Answer:
The text was changed to make it clear that t is the daily time step.
Comment 24:
- Please repeat that “production function” means vertical flow. Add Q(tn) both in the figure caption and in the figure. Explain all abbreviations and give units of variables in the figure. Explain also what is “Upper and lower zone”? The figure needs much clarification.
Answer:
Vertical routing is now stated. Upper zone refers to the unsaturated soil layer. Lower zone refers to groundwater. This is now specified in the text.
Comment 25:
- Equation 2: XKT iis a dimensionless quantity according with the equation. In line 119 it is called the transfer function while in line 122 it is called the routing coefficient. Please provide a clear definition. Subscripts iare missing for variables on the right side of the equation.
Answer:
The sentence was changed to: “CEQUEAU is composed of two main functions, the production function responsible for quantifying the vertical flow of water and the transfer function, which calculates a routing coefficient responsible for quantifying upstream-downstream water advection.”
Comment 26:
- basin: what is the meaning. Is it a whole square or partial square?
Answer:
It is the area of the drainage basin.
Comment 27:
- hydrological simulations: hydrological simulation results.
Answer: The text was corrected.
Comment 28:
- Temperature: Change of temperature.
Answer:
The change was made.
Comment 29:
130-136. Tw is not the temperature but the change of temperature since H is also changing with time. Please give consistent units to all variables. Please add a dot after EXXKT. And replace * with a dot.
Answer:
Tw is the change in temperature when H is the change in heat or enthalpy. The text was changed.
Comment 30:
137-138. Split sample method: Please explain.
Answer:
As stated in the manuscript, split sample indicates that the historical time series used for calibration and validation are split in two sub-series: one used to calibrate the model, and one used to validate the model.
Comment 31:
164-170. Please move these explanations to the first reference to fig. 2.
Answer: Theses sentences were moved after the first reference to Figure 2.
Comment 32:
- Figure 2: The red box on the Canada box is rotated relatively to the large map. CORDEX and NARR must be explained in connection with first reference to fig. 2.
Answer:
The figure was edited. Reanalyses are explained in greater details.
Comment 33:
- GHG: (Green House Gas).
Answer: This was added in the manuscript.
Comment 34:
- RCP 4.5: RCP4.5.
Answer: This mistake was corrected.
Comment 35:
196-197. Please give references.
Answer:
According to our line count, this is the onset of the results section and no references are required.
Comment 36:
- Please indicate the calibration and the evaluation periods in fig. 3.
Answer:
Figure 3 presents the result of a trend analysis on precipitations. Precipitations are not simulated by CEQUEAU. They are an input to the model. Hence the calibration and validation periods are irrelevant.
Comment 37:
209-210. Not shown in fig. 3. Where is it shown?
Answer:
We are not sure what the reviewer is referring to.
Comment 38:
- Figure 2B: Figure 3B.
Answer: The authors mean Figure 3B. This mistake was corrected. The authors thank the reviewer for having pointed out this blunder.
Comment 39:
216-218. Please reformulate this sentence. It is hard to understand especially the meaning of the last part. And what do you mean by patterns.
Answer:
Sentence was reformulated.
Comment 40:
- climate change scenarios: Do you mean the time intervals given in the previous line?
Answer:
Yes climate change scenarios refer to the simulations provided by climate models for the aforementioned periods.
Comment 41:
- This sentence is hard to understand, what is ANOVA?
Answer: The sentence was not well written and it was corrected. The definition of ANOVA was given (Analysis of variance).
Comment 42:
- low flows, which are the focus of the present work: Please explain why this is the focus. Why is it important? And since it says here that it is the focus it should be mentioned much earlier in the paper eventually in the abstract.
Answer:
The beginning of the sentence was changed to: “However, low flows, which are the focus of the present work because of their impact on water availability, water quality and aquatic habitat,”
Comment 43:
- Please indicate calibration and validation periods in the figure.
Answer: [same as comment 36]
Comment 44:
- Note the three figures with A, B, C and also in the figure text.
Answer:
Change was made
Comment 45:
- final set of parameters: What is the meaning of final?
Answer: “Optimal set of parameters” is more meaningful than “final set of parameters” and is used instead in the revised version of the manuscript.
Comment 46:
274-275. Where is the table?
Answer: The authors mean Table 3. This mistake was corrected. The authors thank the reviewer for having pointed out this blunder.
Comment 47:
- decrease by 45-63%.: Please indicate when.
Answer: An explanation is provided in the revised version of the manuscript.
Comment 48:
- Please specify period 1 and 2.
Answer:
Periods were specified.
Comment 49:
- Flow duration: Flow exceedance probability.
Answer:
The terms “Flow duration curve” are often used in hydrology and are accepted.
Comment 50:
- Temperature: change of temperature. Check throughout the paper and replace “temperature” to “change of temperature”.
Answer:
We do not agree. Although the reviewer is right in pointing out that Equation (3) computes and change in temperature if H is a change in enthalpy, the model output is not the relative change in temperature but the actual water temperature at each time step.
Comment 51:
- Remove the last RCP8.5, not necessary to repeat it.
Answer: This mistake was corrected.
Comment 52:
- Change of temperature, not temperature! And add the unit of temperature.
- Slope over time: Please add the unit ◦C / month, ◦C / year ?.
- Month January February August September: Remove this line a little down in the table. It is clear from the table column head.
- Significant trends: Significant trends for temperature change, ◦C.
Answer: unit of the slope was indicated in the Table caption and in the Table.
Comment 53:
- Please explain the meaning of blue boxes, read and black lines.
Answer: An explanation is given in the caption of Figure 8.
Comment 54:
- Please split discussion and conclusion into two separate sections. The conclusion section should be short with a few important results.
Answer:
As stated previously, we decided to keep the sections into one, but it was shortened.
Comment 55:
Please repeat the meaning of PEI.
Answer:
As per any standard scientific text, the acronym was defined at the onset and used subsequently.
Comment 56:
- 100% of: equal to.
Answer:
The text was corrected as suggested by the reviewer.
Comment 57:
- Potential impact: The direct temperature impact on fish are discussed. What about indirect effects such as on nutrients?
Answer:
Given the length of our paper and its focus, we are not discussing nutrients. However, other researchers are focusing on nutrients in PEI and work is underway.
Comment 58:
- Please correct the reference [23] GUY MORIN , DENIS COUILLARD , DANIEL CLUIS , H. GERALD JONES & JEAN-MAURICE GAUTHIER (1987) Prévision des températures de l'eau en rivière à l'aide d'un modèle conceptual, Hydrological Sciences Journal, 32:1, 31-41, DOI: 10.1080/02626668709491160
Answer:
This reference was changed to a reference in English.
Round 2
Reviewer 2 Report
Thank you for detailed responses to my questions and my remarks. I see the Authors did what they can to explain all doubts and improve the text as much as it was possible. In general, I'm satisfied, and I think the manuscript is now ready to be published.